# Predictive Analysis of Postpartum Depression Using Machine Learning

**DOI:** 10.3390/healthcare13080897

**Published:** 2025-04-14

**Authors:** Hyunkyoung Kim

**Affiliations:** Department of Nursing, Kongju National University, Gongju 32588, Republic of Korea; hkk@kongju.ac.kr

**Keywords:** postpartum depression, psychological stress, family conflict, machine learning, pregnant women

## Abstract

**Background:** Maternal postpartum depression (PPD) is a major psychological problem affecting mothers, newborns, and their families after childbirth. This study investigated the factors influencing maternal PPD and developed a predictive model using machine learning. **Methods/Design:** In this study, we applied machine learning techniques to identify significant predictors of PPD and to develop a model for classifying individuals at risk. Data from 2570 subjects were analyzed using the Korean Early Childhood Education and Care Panel (K-ECEC-P) dataset as of January 2025, utilizing Python version 3.12.8. **Results:** We compared the performance of a decision tree classifier, random forest classifier, AdaBoost classifier, and logistic regression model using metrics such as precision, accuracy, recall, F1-score, and area under the curve. The logistic regression model was selected as the best model. Among the 13 features analyzed, conflict with a partner, stress, and the value of children emerged as significant predictors of PPD. **Discussion:** Conflict with a partner and stress levels emerged as the strongest predictors. Higher levels of conflict and stress were associated with an increased likelihood of PPD, whereas a higher value of children reduced this risk. Maternal psychological status and environmental features should be managed carefully during the postpartum period.

## 1. Introduction

Postpartum depression (PPD) has become more common in recent years, partly due to the pandemic and its associated challenges, such as heightened stress, social isolation, limited access to healthcare services, and financial instability [1]. The prevalence of PPD doubled, increasing from 9.4% in 2010 to 19.0% in 2021. PPD rates increased across all racial and ethnic groups, with the highest relative growth observed among Asians and Pacific Islanders, rising from 3.6% in 2010 to 13.8% in 2021 [2]. Various risk factors have compounded the emotional and physical strains of new motherhood, contributing to the rising rates of PPD [1].

Recent meta-analyses have found that PPD is influenced by various demographic, medical, psychological, social, and instrumental factors [3,4]. Demographic factors, including young maternal age, lower education levels, low socioeconomic status, and unemployment, increase the risk of PPD [3,4]. Medical factors, such as gestational diabetes, pregnancy complications, cesarean delivery, the use of epidural anesthesia, and infant health issues (e.g., low birth weight, intensive care unit admission), are also significant predictors [3,4,5]. Psychological risk factors, including a history of depression or anxiety, high neuroticism, emotional instability, and trauma, further contribute to PPD vulnerability [3,4]. Among social factors, inadequate support from family [3], female friends, or one’s mother [6] has been reported to increase PPD risk. Lastly, instrumental factors, including limited healthcare access, lack of mental health screening, and unhealthy lifestyles, further exacerbate the risk of PPD [3,4].

However, the most influential factor remains unclear because several studies have reported different variables. For example, one study reported that the husband’s low level of education, low social support, vacuum use during delivery, and stressful life events during pregnancy were the most significant factors affecting PPD [5]. Another study identified a wide range of relevant factors, including immigration challenges, obesity or dissatisfaction with body image, vitamin D deficiency, genetic and epigenetic markers, low self-esteem, multiple births, postpartum anemia, cultural preferences for male children, and social stigma surrounding mental health [7].

Machine learning techniques may help address these challenges by identifying significant features that contribute to predicting health outcomes. In large samples, machine learning serves as a powerful computational tool for analyzing multivariate medical data. By integrating a multitude of parameters, machine learning can offer a more objective assessment of PPD risk, potentially enabling early interventions and improved outcomes [8]. Machine learning models have been applied to predict PPD using diverse datasets, including electronic health records (EHRs) and patient-reported outcomes. These models identified key predictive factors across demographic (age, marital status, and socioeconomic status) [9,10,11,12], medical (pregnancy complications, body mass index, and medication) [9,13,14,15,16], psychological (neuroticism, resilience, and emotional stability) [10,11,17], social (number of people living together) [18], and instrumental (healthcare access and prenatal care) [12,15,17] domains. Among these, mental health history, pregnancy complications, and socioeconomic conditions were the strongest predictors of PPD risk [9,10,14,15]. Machine learning models have highlighted the significant influence of demographic and medical factors in predicting the risk of PPD. However, there remains a lack of machine learning analysis of social aspects, individual emotional characteristics, and health behaviors.

Therefore, this study attempts to use machine learning analytics to integrate individual health behaviors, social support, emotional characteristics, demographic variables, and psychological characteristics from large datasets. By integrating numerous parameters, machine learning can offer a more objective assessment of PPD risk, potentially enabling early interventions and improved outcomes. Understanding PPD risk factors will be crucial for early detection, prevention, and targeted interventions in maternal healthcare. This study aimed to identify significant predictors of PPD and develop a predictive model to classify individuals at risk using machine learning.

## 2. Methods and Materials

### Study Design

This study employed a quantitative, cross-sectional design to analyze public data from the Korean Early Childhood Education and Care Panel (K-ECEC-P) using an artificial intelligence-based machine learning model. The cross-sectional approach facilitated a snapshot analysis of key associations with PPD in a large, representative sample of postpartum women. In large samples, machine learning serves as a powerful computational tool for analyzing multivariate medical data [8].

Machine learning was applied in this study as an advanced analytical tool to improve the accuracy of PPD risk prediction, an area where traditional statistical models have limitations. Unlike classical regression techniques, which rely on predefined assumptions about the relationships among variables, machine learning models can analyze complex, non-linear interactions among multiple predictors, uncovering hidden patterns that may not be apparent in traditional analyses. The use of algorithms such as random forest, AdaBoost, and logistic regression enabled a comparative assessment of classification performance, ensuring that the most effective model was selected based on empirical metrics such as accuracy, precision, recall, and area under the receiver operating characteristic curve (AUC-ROC) scores.

Machine learning logistic regression and general logistic regression differ in purpose, methodology, and applications. General logistic regression is primarily used for inferential analysis and hypothesis testing, focusing on understanding relationships between variables. In contrast, machine learning logistic regression is designed to optimize performance in predictive modeling and classification. Additionally, machine learning logistic regression scales better for high-dimensional and large datasets, whereas general logistic regression struggles with complexity and overfitting [18].

## 3. Materials

The K-ECEC-P is a longitudinal research project that tracks the growth and development of children from the prenatal stage through elementary school, conducted by the Korea Institute of Child Care and Education (KICCE). This study utilized one-month postpartum maternal panel data from 2022, which were retrieved on 2 December 2024 [19]. The K-ECEC-P applies Bronfenbrenner’s ecological systems theory [20] and Elder’s life course theory [21] to track environmental variables that influence children’s healthy development. It examines the dynamic relationships between children and the multifaceted environments surrounding them, including their families. The one-month postpartum maternal survey was administered online to postpartum women who had completed the baseline questionnaire during pregnancy. The survey was conducted between February and September 2022, according to the expected delivery dates provided during the baseline survey by trained research assistants. KICCE handled data editing, cleaning, weighting, supplementation, and adjustments for missing data or non-responses. KICCE also developed the survey items along with a codebook, profile, and user guide. The data were released to panel data users from October to November 2024.

### Sampling

Systematic sampling was used for the K-ECEC-P, which involved sorting medical institutions within each stratum based on regional codes. The target population was defined, and subpopulation proportions were identified using census data. The sampling unit and frame were finalized, the sample size and target error level determined, and stratification and sampling executed, with weights calculated and a parameter estimation method established. The final recruitment size for pregnant women was 3380, with the following regional distribution: Seoul 521 (15.4%), Gyeonggi/Incheon 1037 (30.7%), Daejeon/Sejong/Chungcheong/Gangwon 365 (10.8%), Daegu/Gyeongbuk 409 (12.1%), Busan/Ulsan/Gyeongnam 572 (16.9%), and Gwangju/Jeolla/Jeju 476 (14.1%). Among the 3380 recruited pregnant women, 95 (2.8%) were from multicultural households, 72 (2.1%) were from low-income households, and 62 (1.8%) had twin pregnancies. This study used the full dataset, comprising 2790 respondents out of 3133 survey participants, representing an 89.1% response rate.

## 4. Features

### 4.1. General Characteristics

The features included personal characteristics such as age, education, marital status, and maternal weight. Maternal age (in years), education (categorized as elementary, middle, high school, college, master’s, or doctoral degree), marital status (married or unmarried), and maternal weight (in kilograms) were self-reported.

### 4.2. Birth Related Characteristics

The features also included the number of children, the use of a nursery center after birth (yes or no), type of birth (normal spontaneous vaginal birth, elective cesarean section, or emergency cesarean section), type of feeding (breastfeeding, bottle feeding, or mixed feeding), and first feeding time (hours after birth), all of which were self-reported.

### 4.3. Conflict

Conflict was measured using the Conflict **with** Partner Scale, developed by Jung et al. [22] based on Markman et al.’s study [23]. The scale comprises eight items rated on a 5-point Likert scale ranging from “Not at all true” (1 point) to “Very true” (5 points), with higher scores indicating a higher level of spousal conflict. The scale demonstrated a Cronbach’s alpha of 0.96 in the original study and 0.93 in this study.

### 4.4. Stress

Stress was assessed using the Parenting Stress Scale (PSS) [24]. The PSS contains 10 items rated on a 5-point Likert scale from “Not at all true” (1 point) to “Very true” (5 points), with higher scores indicating greater parental pressures and distress. The scale demonstrated a Cronbach’s alpha of 0.88 in the original study and 0.87 in this study.

### 4.5. Value of Children

This variable was measured using the Value of Children Scale [25]. The scale consists of eight items divided into two subdomains—emotional value (items 1–4) and instrumental value (items 5–8)—and is rated on a 5-point Likert scale ranging from “Not at all true” (1 point) to “Very true” (5 points). Higher scores indicate that parents attribute greater emotional and instrumental value to their children. The scale demonstrated a Cronbach’s alpha of 0.88 in the original study and 0.78 in this study.

### 4.6. Paternal Participation

Paternal participation was assessed using the Father’s Role Perception Scale [26]. This scale comprises four items rated on a 5-point Likert scale from “Not at all true” (1 point) to “Very true” (5 points), with higher scores indicating greater father involvement in child-rearing. The scale exhibited a Cronbach’s alpha of 0.82 in the original study and 0.77 in this study.

## 5. Label

### 5.1. PPD

The target variable, PPD, was measured using the Korean version of the Edinburgh Postnatal Depression Scale (K-EPDS) [27], which is a validated adaptation of the 10-item EPDS [28]. A cutoff score of 9.5 was recommended to identify both major and minor depression, taking into account sensitivity, specificity, and positive predictive value [27]. The scale comprises 10 items rated on a 4-point Likert scale (0–3 points). After reverse-coding all items except for items 1, 2, and 4, the total score is obtained by summing the responses. Higher scores indicate more severe depressive symptoms. Scores from 0 to 9 denote the non-depressed group, while scores from 10 to 30 indicate the depressed group. For data pre-processing, the PPD groups were dichotomized as 0 and 1. The scale demonstrated a Cronbach’s alpha of 0.85 in the original study and 0.74 in this study.

### 5.2. Ethical Considerations

This study was approved by the Institutional Review Board of KICCE (No. KICCEIRB-2022-01) and adhered to the Declaration of Helsinki. Informed consent was obtained from all participants in the study.

## 6. Analysis

A supervised machine learning model was used to predict PPD. The analysis was performed using Python version 3.12.8. The dataset comprised 2570 entries and 13 features, including age, marital status, education level, maternal weight, stress, value of children, conflict, paternal participation, type of birth, feeding type, first feeding time, number of children, and use of a nursery, along with a target outcome indicating depression status (1 = PPD, 0 = no PPD). Pre-processing involved standardization and data overview, with z-score normalization applied to ensure that features were on a similar scale. No missing values were detected; however, outliers were analyzed for their impact on model performance and were capped using an interquartile range-based method for extreme values in age and maternal weight. To address class imbalance (with 75.95% of entries representing PPD cases and 24.05% representing non-PPD cases). The dataset was split into training and test sets at an 8:2 ratio. For model selection, four machine learning models—decision tree classifier, random forest classifier, AdaBoost classifier, and logistic regression—were evaluated using metrics including accuracy, precision, recall, F1-score, and area under the receiver operating characteristic curve (AUC-ROC). ROC-AUC is a performance metric for binary classification models that evaluates their ability to distinguish between classes. Accuracy is defined as the proportion of correctly classified instances out of the total predictions, providing a general measure of a model’s performance. Precision refers to the proportion of correctly predicted positive instances among all predicted positives, which is particularly important in cases where false positives are costly. Recall represents the proportion of actual positive instances correctly identified by the model, which is crucial when minimizing false negatives is a priority. The F1-score is the harmonic mean of precision and recall, serving as a balanced metric for evaluating models, especially in imbalanced datasets where both false positives and false negatives need to be considered [29].

## 7. Results

### 7.1. Data Characteristics

The study population had a mean maternal age of 33.52 years, an average marriage duration of 49.2 months, and an average body weight of 68.05 kg. Most mothers practiced breastfeeding (77.0%), while 39.2% used bottle feeding. Vaginal births accounted for 38.6% of deliveries, cesarean sections for 40.8%, and emergency cesarean sections for 20.6%. The average number of children was 3.26, with a range of 1 to 7. In terms of education, 69.9% of participants had completed high school. The average partner conflict score was 15.45, and the mean scores for paternal participation in rearing and value of children were 3.87 and 25.84, respectively. PPD was highly prevalent, affecting 24.05% of mothers, with a mean EPDS score of 15.17 (see Table 1).

### 7.2. Model Performance Comparison

A comparative evaluation of four machine learning models—decision tree, random forest, AdaBoost, and logistic regression—was conducted to assess classification performance. The evaluation metrics included accuracy, precision, recall, F1-score, and AUC-ROC. The random forest model achieved the highest accuracy (77.04%), followed closely by AdaBoost (76.85%) and logistic regression (75.49%). The decision tree model had the lowest accuracy (66.54%), indicating limited generalization. Although the decision tree model exhibited the highest precision (79.95%), indicating fewer false positives, its lower recall (74.62%) suggests it struggled to identify all positive cases. In contrast, random forest (97.44%) and logistic regression (97.18%) achieved the highest recall, meaning they were more effective in capturing positive instances, while AdaBoost performed well with a recall of 96.41%. The random forest model attained the highest F1-score (86.56%), demonstrating a strong balance between precision and recall, with AdaBoost (86.34%) and logistic regression (85.75%) following closely; the decision tree’s F1-score was notably lower at 77.19%. Furthermore, the decision tree model had the lowest AUC-ROC (0.58), indicating weak overall classification performance, whereas logistic regression achieved the highest AUC-ROC (0.74), with AdaBoost and random forest closely behind at 0.73. Overall, while random forest, AdaBoost, and logistic regression performed similarly in terms of accuracy, recall, and F1-score, logistic regression’s superior AUC-ROC suggests it may be the best model for scenarios requiring strong overall discrimination (see Table 2 and Figure 1).

### 7.3. Feature Importance from Logistic Regression

The logistic regression model identified conflict, stress, value of children, age, and education as the most influential predictors of PPD. Specifically, conflict (coefficient 0.50) and stress (0.39) showed positive correlations, indicating that higher levels of these factors were associated with an increased likelihood of PPD. In contrast, the value of children (coefficient −0.14) had a negative correlation, suggesting that a higher value of children was associated with a decreased risk. Age (0.10) and education (0.09) also exhibited positive correlations (see Table 3 and Figure 2).

## 8. Discussion

This study developed a machine learning-based model to predict PPD and identified key psychological and social factors. The results indicate that conflict with a partner, stress, and the value of children significantly influence PPD risk. These findings are consistent with prior research [30] that emphasizes the importance of mental and emotional well-being during the postpartum period. Higher levels of conflict with a partner were found to significantly increase the likelihood of PPD, and previous studies have demonstrated that a lack of emotional and physical support from a spouse after childbirth can destabilize maternal emotions and exacerbate depressive symptoms [30]. Moreover, the combination of high parenting stress further elevates the risk of PPD [31]. Overall, these findings underscore the importance of emotional and practical support from spouses and family members in maintaining postpartum mental health.

In addition, the study found that the value of children significantly related to the occurrence of PPD. This represents a novel analytical finding that has rarely been explored in previous machine learning–based PPD studies [31]. The results indicate that a higher perceived value of children is associated with a reduced risk of PPD. In contrast, mothers with PPD may struggle to bond emotionally with their infants, displaying less warmth, affection, and joy due to emotional distress [32]. This suggests that a positive perception regarding the value of children can serve as a protective factor against PPD, emphasizing the need for future research to examine how this factor interacts with cultural influences. Consequently, psychological assessments exploring postpartum women’s perceptions of their children should be incorporated into PPD prevention strategies.

Most existing machine learning studies have focused on factors such as mental health history [30], delivery methods (including cesarean section and medication use during birth) [31], socioeconomic factors [32,33], health behaviors (such as smoking and sleep) [8,30], and birth complications [34]. Fewer studies have thoroughly examined the impact of spousal and family relationships on PPD. Additionally, many current machine learning studies [30,31,32,33,34,35] have concentrated on mental health records, socioeconomic factors, and birth data. In this study, however, age, education, number of children, feeding method, use of a nursery, marriage duration, paternal participation, cesarean section, first feeding time, and maternal weight were not statistically significant predictors. Previous studies have identified personal factors such as maternal age under 25 [8], low educational attainment [35], and high antenatal BMI [8,34] as significant predictors of PPD. Furthermore, institutional factors like social support [30] and healthcare service utilization [34] have been shown to mitigate PPD risk. The contrast between these findings and the present study suggests that influencing factors may vary across different populations or study designs. Therefore, future models should incorporate a broader range of predictors—including personal, institutional, biological, and hormonal factors—to improve PPD prediction.

In this study, logistic regression demonstrated the best performance with an AUC of 0.74. Most machine learning-based PPD prediction models report AUC values ranging from 0.7 to 0.9. For instance, self-reported survey-based models in patients with immune-mediated inflammatory disease have shown AUCs between 0.87 and 0.91 [36], while a PPD cohort study reported an AUC of 0.71 [34]. Electronic health record-based machine learning models have achieved average AUCs of 0.72 to 0.84 [8]. Logistic regression is a supervised learning method that predicts the probability of an outcome using a linear combination of independent variables [37]. A decision tree is a supervised learning algorithm used for both classification and regression tasks. It mimics human decision-making by partitioning a dataset into smaller subsets using if–then conditions, forming a tree-like structure [37]. AdaBoost is an ensemble learning technique that improves weak classifiers by sequentially adjusting their weights to focus on difficult-to-classify samples [38]. A random forest model constructs bootstrap samples by selecting n data points from the training dataset and randomly choosing subsets of input variables to generate multiple decision trees [39]. Each model offers unique advantages; the choice depends on whether the priority is interpretability, accuracy, or computational efficiency. In this study, logistic regression (AUC-ROC 0.74) was most suitable for clinical interpretation, while random forest and AdaBoost (both AUC-ROC 0.73) were more appropriate for high-accuracy prediction tasks. The decision tree model, with the lowest AUC-ROC (0.58), appears less suitable for complex cases.

Based on the study results, several strategies are necessary to predict and prevent PPD. First, healthcare providers and policymakers should enhance emotional support programs for postpartum women and families, including couples counseling and family support services for high-risk groups. These findings also suggest that interventions aimed at improving partner support and reinforcing positive maternal identity may help reduce the burden of PPD. By identifying these relationships, this study underscores the importance of considering both partnership dynamics and maternal values in the assessment and prevention of PPD. Second, researchers and mental health professionals should implement prenatal and early postpartum psychological assessments—within four weeks after birth—that evaluate spousal relationships, stress levels, and parental values. Future research should further explore these associations across diverse populations and cultural contexts to better understand how social and psychological factors interact in influencing PPD risk [6]. Third, clinicians and healthcare institutions should expand the clinical application of PPD prediction models by integrating maternal value and paternal support level with electronic health record data to facilitate early intervention. Fourth, global health organizations and academic institutions should conduct multinational studies that account for cultural differences, as the relationship between parental values and PPD may vary across cultures. Lastly, data scientists and medical researchers should develop integrated machine learning models that combine physiological, environmental, and psychological factors for more precise PPD prediction in future research.

This study has several limitations. Machine learning models—particularly complex ones like random forest or AdaBoost—are often considered “black box” models due to their lack of interpretability. This limitation is especially challenging in sensitive domains such as healthcare and PPD prediction, where unexplainable predictions may lead to biased or incorrect results without clear avenues for correction. Additionally, class imbalance and a limited dataset may affect the model’s generalizability. Another limitation was related to the psychological predictors used in the PPD. The analysis focused on stress, conflict with a partner, and the value of children, but did not include other significant psychosocial factors. Key predictors such as emotional resilience, coping mechanisms, personality traits (e.g., neuroticism), and perceived social support were absent, limiting the model’s comprehensiveness. Furthermore, the study did not include various forms of social support, such as religious organizations, community networks, or peer groups, which are not known to influence PPD risk but may have impacted the model’s predictive accuracy. The reliance on self-reported data introduces potential biases, as responses may have been influenced by stigma or subjective perceptions. Moreover, the study did not explore how these psychological factors interact with each other or moderate the risk of PPD, reducing the depth of analysis. Future research should address these gaps by incorporating a broader range of social variables, integrating medical history data, and conducting cross-cultural comparisons to enhance predictive accuracy and clinical relevance. The dataset did not include certain known risk factors, such as socioeconomic status and a history of mental illness, and electronic health records were not incorporated, thereby limiting the use of medical history-based variables. Furthermore, as the study was conducted on a Korean population, further research is needed to assess its applicability to different cultural contexts.

## 9. Conclusions

Machine learning offers valuable insights into the factors influencing postpartum depression. By identifying key predictors such as conflict, stress, and the value of children, this study lays the foundation for developing targeted interventions to mitigate PPD risk. This study provides important insights into the relationship between partner social support and PPD, an area that has been insufficiently addressed in both classical statistical and machine learning analyses. The findings indicate that partner-related factors, including conflict with a partner and paternal involvement, are significantly associated with PPD risk. Higher levels of partner conflict were linked to an increased likelihood of PPD, while greater paternal involvement was associated with lower depressive symptoms. These results suggest that the quality of partner support during the perinatal period plays a crucial role in maternal mental health. Additionally, this study highlights a novel association between maternal social values and beliefs about children and PPD risk. Mothers who placed a higher value on children exhibited lower levels of PPD symptoms, suggesting that positive perceptions of motherhood may have a protective effect on mental well-being. This finding emphasizes the potential role of maternal attitudes and societal beliefs in shaping postpartum emotional experiences. Further refinement of these models and the inclusion of additional data will enhance predictive accuracy and practical applications.

## Figures and Tables

**Figure 1 healthcare-13-00897-f001:**
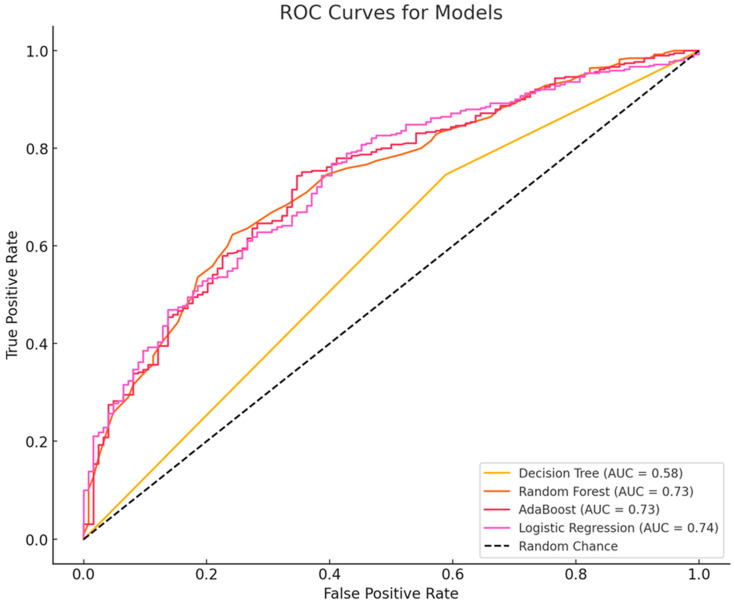
Receiver operating characteristic (ROC) curve analysis.

**Figure 2 healthcare-13-00897-f002:**
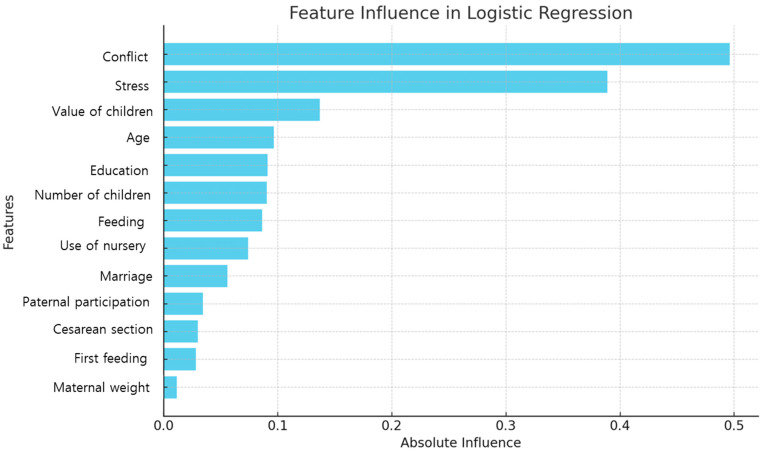
Feature importance based on the logistic regression.

**Table 1 healthcare-13-00897-t001:** Characteristics of datasets (n = 2570).

Variables	Categories	n (%)	M ± SD	Min–Max
Age (year)			33.52 ± 4.29	17–49
Maternal body weight (kg)			68.05 ± 12.16	37–125
Marriage period (months)			49.20 ± 32.57	6–245
Feeding	Bottle feeding	1980 (77.0)		
	Breastfeeding	590 (39.2)		
First feeding time from birth (hours)			6.02 ± 1.92	1–9
Number of children			3.26 ± 1.12	1–7
Type of birth	Vaginal birth	992 (38.6)		
	Cesarean section	1049 (40.8)		
	Emergency section	529 (20.6)		
Use of nursery	Yes	2215 (86.2)		
	No	355 (13.8)		
Education	Elementary school	2 (0.1)		
	Middle school	263 (10.2)		
	High school	1796 (69.9)		
	College	294 (11.4)		
	Master or doctor	19 (0.7)		
	Not reported	166 (6.5)		
Conflict with partner			15.45 ± 7.24	8–72
Stress			25.67 ± 7.38	10–49
Paternal participation in rearing			3.87 ± 3.12	4–20
Value of children			25.84 ± 4.68	9–40
Postpartum depression (EPDS)			15.17 ± 4.51	1–30
EPDS	Non-PPD (0–9)	2164 (75.95)		
	PPD (10–30)	473 (24.05)		

EPDS = Edinburgh Postnatal Depression Scale, PPD = postpartum depression.

**Table 2 healthcare-13-00897-t002:** Machine learning model performance comparison (n = 2570).

Model	Accuracy	Precision	Recall	F1-Score	AUC-ROC
Decision Tree	66.54%	79.95%	74.62%	77.19%	0.58
Random Forest	77.04%	77.87%	97.44%	86.56%	0.73
AdaBoost	76.85%	78.17%	96.41%	86.34%	0.73
Logistic Regression	75.49%	76.72%	97.18%	85.75%	0.74

AUC-ROC = Area Under the Curve—Receiver Operating Characteristic.

**Table 3 healthcare-13-00897-t003:** Feature importance from logistic regression (n = 2570).

Feature	Coefficient	Absolute Influence	OR	*p*
Conflict	0.50	0.50	1.68	<0.001
Stress	0.39	0.39	1.61	<0.001
Value of children	−0.14	0.14	0.87	0.018
Age	0.10	0.10	1.10	0.126
Education	0.09	0.09	1.10	0.116
Number of children	−0.09	0.09	1.09	0.113
Feeding	−0.09	0.09	0.92	0.135
Use of nursery	0.07	0.07	1.08	0.185
Marriage	−0.06	0.06	0.95	0.367
Paternal participation	−0.03	0.03	0.97	0.556
Cesarean section	−0.03	0.03	0.97	0.607
First feeding	0.03	0.03	1.03	0.626
Maternal weight	0.01	0.01	1.01	0.830

Absolute value of coefficients = the average of the magnitudes of all feature coefficients (ignoring their signs). This metric can be used to summarize the overall “strength” of the relationship between features and the target variable across the model. OR = Odds Ratio.

## Data Availability

Please contact the corresponding author for data availability.

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
