# Peer review of "Predictive Analysis of Postpartum Depression Using Machine Learning"

_healthcare, 2025, doi:10.3390/healthcare13080897_

Round 1
Reviewer 1 Report
Comments and Suggestions for Authors
Title: Predictive Analysis of Postpartum Depression Using 2 Machine Learning
Please amend some of the terms utilised in the demographic background section- there is mention of ‘Whites’ ‘Hispanic’ and ‘Blacks’. It is important to be ethically sensitive to ensure no offence for the reader; for instance, what type of ‘black’ individuals- Black Caribbean, African American, British, Korean?
Line 69: Please make an appropriate in-text reference citation and remove the hyperlink.
This study is very interesting and adds machine learning to identify factors and predict PPD. Several key predictors were identified surrounding conflict with partners, stress and value for children. It was rightly pointed out that healthcare providers should offer emotional support programs, counselling and implement psychological assessments.
The methodological structure, results, findings and limitations are clearly demonstrated. The use of a predictive analysis shown promising results that future research may adapt in other international studies.
Author Response
Please amend some of the terms utilised in the demographic background section- there is mention of ‘Whites’ ‘Hispanic’ and ‘Blacks’. It is important to be ethically sensitive to ensure no offence for the reader; for instance, what type of ‘black’ individuals- Black Caribbean, African American, British, Korean?
2p: The prevalence of PPD doubled, increasing from 9.4% in 2010 to 19.0% in 2021. PPD rates increased across all racial and ethnic groups, with the highest relative growth observed among Asians and Pacific Islanders, rising from 3.6% in 2010 to 13.8% in 2021 [2].
Line 69: Please make an appropriate in-text reference citation and remove the hyperlink.
4p: The cross-sectional approach facilitated a snapshot analysis of key associations with PPD in a large, representative sample of postpartum women. In large samples, machine learning serves as a powerful computational tool for analyzing multivariate medical data [8].
This study is very interesting and adds machine learning to identify factors and predict PPD. Several key predictors were identified surrounding conflict with partners, stress and value for children. It was rightly pointed out that healthcare providers should offer emotional support programs, counselling and implement psychological assessments.
16p: These results suggest that the quality of partner support during the perinatal period plays a crucial role in maternal mental health. Additionally, this study highlights a novel association between maternal social values and beliefs about children and PPD risk. Mothers who placed a higher value on children exhibited lower levels of PPD symptoms, suggesting that positive perceptions of motherhood may have a protective effect on mental well-being. This finding emphasizes the potential role of maternal attitudes and societal beliefs in shaping postpartum emotional experiences.
The methodological structure, results, findings and limitations are clearly demonstrated. The use of a predictive analysis shown promising results that future research may adapt in other international studies.
Thank you for your constructive suggestions.
Reviewer 2 Report
Comments and Suggestions for Authors
Verifying predictors of postpartum depression using machine learning provides some clinically useful information and knowledge. However, some aspects of the writing were found to need to be revised. The rationale for the study design, research gaps, and findings should be further discussed. Here's what seems like it needs improvement:
1. In the introduction, you need to describe more clearly why you are doing this study and the rationale. Please describe in detail the research on factors related to Postpartum Depression and the models that can predict it so far, and provide more specific evidence that machine learning can effectively predict Postpartum Depression.
2. At the end of the introduction, you stated that this study contributes to predicting health outcomes, but you need to specifically state what this study will add to the findings of previous studies. In other words, you need to describe how this study can fill the research gap. If you do so, you will need to review more relevant studies and include appropriate references.
3. It is recommended that the subtitles of the manuscript be in the following order: Introduction, Methods and Materials, Results, Discussion, and Conclusion. Measurements, Analysis, etc. are included under Methods and Materials.
4. Using machine learning to perform logistic regression analysis may be a good choice. However, can the results obtained through this method be differentiated from the results of logistic regression analysis studied in previous research?
5. Perhaps because of the limitations in the included variables, the derived results lack detailed variable content other than general content and do not appear to have any outstanding findings. This issue should also be presented as a limitation of the study.
6. The limitation of this study is that there are almost no psychosocial variables as predictors. Please present this as a limitation of the study and provide suggestions for future research.
7. It would be nice to be a little more specific about the clinical implications of what we found in this study. Please provide specific information and knowledge on how to prevent PPD as revealed in the findings of this study, and what should be considered when treating PPD.
8. It would be nice to be a little more specific about the conclusions. As I mentioned earlier, there doesn't seem to be any outstanding findings, but the conclusion is too simple. If you consider the clinically meaningful findings among the results and highlight their significance, I think you can describe the conclusion more specifically.
Author Response
Verifying predictors of postpartum depression using machine learning provides some clinically useful information and knowledge. However, some aspects of the writing were found to need to be revised. The rationale for the study design, research gaps, and findings should be further discussed. Here's what seems like it needs improvement:
I appreciate your thoughtful review and the opportunity to strengthen this work based on your feedback.
- In the introduction, you need to describe more clearly why you are doing this study and the rationale. Please describe in detail the research on factors related to Postpartum Depression and the models that can predict it so far, and provide more specific evidence that machine learning can effectively predict Postpartum Depression.
3p: Machine learning models have been applied to predict PPD using diverse datasets, including electronic health records (EHRs) and patient-reported outcomes. These models identified key predictive factors across demographic (age, marital status, and socioeconomic status) [9-12], medical (pregnancy complications, body mass index, and medication) [9,13-16], psychological (neuroticism, resilience, emotional stability) [10,11,17], social (number of people living together) [18], and instrumental (healthcare access and prenatal care) [12,15,17] domains. Among these, mental health history, pregnancy complications, and socioeconomic conditions were the strongest predictors of PPD risk [9,10,14,15]. Machine learning models have highlighted the significant influence of demographic and medical factors in predicting the risk of PPD. However, there remains a lack of machine learning analysis of social aspects, individual emotional characteristics, and health behaviors.
- At the end of the introduction, you stated that this study contributes to predicting health outcomes, but you need to specifically state what this study will add to the findings of previous studies. In other words, you need to describe how this study can fill the research gap. If you do so, you will need to review more relevant studies and include appropriate references.
3-4p: Therefore, this study attempts to use machine learning analytics to integrate individual health behaviors, social support, emotional characteristics, demographic variables, and psychological characteristics from large datasets. By integrating numerous parameters, machine learning can offer a more objective assessment of PPD risk, potentially enabling early interventions and improved outcomes. Understanding PPD risk factors will be crucial for early detection, prevention, and targeted interventions in maternal healthcare.
- It is recommended that the subtitles of the manuscript be in the following order: Introduction, Methods and Materials, Results, Discussion, and Conclusion. Measurements, Analysis, etc. are included under Methods and Materials.
Introduction, Methods and Materials, Results, Discussion, and Conclusion.
- Using machine learning to perform logistic regression analysis may be a good choice. However, can the results obtained through this method be differentiated from the results of logistic regression analysis studied in previous research?
2p; Recent meta-analyses have found that PPD is influenced by various demographic, medical, psychological, social, and instrumental factors [3,4]. Demographic factors, including young maternal age, lower education levels, low socioeconomic status, and unemployment, increase the risk of PPD [3,4]. Medical factors, such as gestational diabetes, pregnancy complications, cesarean delivery, the use of epidural anesthesia, and infant health issues (e.g., low birth weight, intensive care unit admission), are also significant predictors [3,4]. Psychological risk factors, including history of depression or anxiety, high neuroticism, emotional instability, and trauma, further contribute to PPD vulnerability [3,4]. Among social factors, inadequate support from family [3], female friends, or one’s mother [6] has been reported to increase PPD risk. Lastly, instrumental factors, including limited healthcare access, lack of mental health screening, and unhealthy lifestyles, further exacerbate the risk of PPD [3,4].
- Perhaps because of the limitations in the included variables, the derived results lack detailed variable content other than general content and do not appear to have any outstanding findings. This issue should also be presented as a limitation of the study.
15p: Another limitation was related to the psychological predictors used in the PPD. The analysis focused on stress, conflict with a partner, and the value of children, but did not include other significant psychosocial factors. Key predictors such as emotional resilience, coping mechanisms, personality traits (e.g., neuroticism), and perceived social support were absent, limiting the model's comprehensiveness. Furthermore, the study did not include various forms of social support, such as religious organizations, community networks, or peer groups, which are not known to influence PPD risk but may have impacted the model's predictive accuracy.
- The limitation of this study is that there are almost no psychosocial variables as predictors. Please present this as a limitation of the study and provide suggestions for future research.
15-16p: The reliance on self-reported data introduces potential biases, as responses may have been influenced by stigma or subjective perceptions. Moreover, the study did not explore how these psychological factors interact with each other or moderate the risk of PPD, reducing the depth of analysis. Future research should address these gaps by incorporating a broader range of social variables, integrating medical history data, and conducting cross-cultural comparisons to enhance predictive accuracy and clinical relevance.
- It would be nice to be a little more specific about the clinical implications of what we found in this study. Please provide specific information and knowledge on how to prevent PPD as revealed in the findings of this study, and what should be considered when treating PPD.
16p: This study provides important insights into the relationship between partner social support and PPD, an area that has been insufficiently addressed in both classical statistical and machine learning analyses. The findings indicate that partner-related factors, including conflict with a partner and paternal involvement, are significantly associated with PPD risk. Higher levels of partner conflict were linked to an increased likelihood of PPD, while greater paternal involvement was associated with lower depressive symptoms. These results suggest that the quality of partner support during the perinatal period plays a crucial role in maternal mental health.
- It would be nice to be a little more specific about the conclusions. As I mentioned earlier, there doesn't seem to be any outstanding findings, but the conclusion is too simple. If you consider the clinically meaningful findings among the results and highlight their significance, I think you can describe the conclusion more specifically.
16-17p: Additionally, this study highlights a novel association between maternal social values and beliefs about children and PPD risk. Mothers who placed a higher value on children exhibited lower levels of PPD symptoms, suggesting that positive perceptions of motherhood may have a protective effect on mental well-being. This finding emphasizes the potential role of maternal attitudes and societal beliefs in shaping postpartum emotional experiences.
Reviewer 3 Report
Comments and Suggestions for Authors
This is an impressive and innovative study. It needs significant attention to the following issues.
1. The Introduction is quite short. There is a good deal of research on this subject that is not cited. I'd suggest that another 10-12 references be included. Some attention may be given to the role of social supports on PPD (embeddedness in networks such as extended family, community groups, congregations, etc.). Even if the influence of these support networks cannot be measured due to data limitations (see comment 2), this is worth mentioning at the outset.
2. Does the data set feature social support variables such as those listed in my foregoing comment? That is a missing component here. If not, please circle back to that limitation in the Discussion and recommend that as a path for future research.
3. A Methods section of one sentence seems really limited. Could this section be expanded to at least 2-3 paragraphs that delve more deeply into the methods applied and their value? This is an opportunity to make a strong case for your design and it is a missed opportunity. As one example, you state the following in the Introduction: "By integrating a multitude of parameters, machine learning can offer a more objective assessment of PPD risk, potentially enabling early interventions and improved outcomes
[6]." A statement like this could be expanded on in the Methods section. How does machine learning offer a more objective assessment? Would this approach expected to be superior to more conventional methods such as regression? Unpack the logic and potential advantages of machine learning approaches for those unfamiliar with them. Moreover, since there are multiple machine learning approaches, detailing those variations and their relative potential contributions seems appropriate here. One sentence is simply too terse.
4. I have always thought that the Value for Children scale should be renamed the Value of Children scale, but I expect no name change here. You might add that it is sometimes called the Children's Perceived Value scale. It is my understanding that these names are used interchangeably.
5. Breastfeeding should be one word. It is sometimes one word (text) and other times two words (tables).
6. I'd welcome the addition of citations to support the use of SMOTE, AUC-ROC, and any analytical approach employed here. I will readily admit that I am not an expert on these methods. Yet, their use should be justified with citations to relevant peer-reviewed publications. Be sure that each is sufficiently explained without presuming prior knowledge on the part of the reader.
7. If I understand correctly, the author used a cross-section of data from a panel study. We cannot then say that Value for Children has an effect or influence on PPD. The causal direction here is impossible to determine from a single wave of data. Value for Children is a dynamic construct that could influence or by influenced by PPD. So, an argument that Value for Children "influences the occurrence of PPD" should be "is associated with the occurrence of PPD." Special care should be taken to re-read the entire manuscript with causal order considerations and data limitations in mind. If the data analyzed here are longitudinal, please state that more clearly.
8. What broader conclusions about machine-learning approaches versus regression models can be discerned from this manuscript?
This is a promising study but needs considerable additional work to realize its potential.
Author Response
This is an impressive and innovative study. It needs significant attention to the following issues.
I sincerely thank the reviewer for your valuable time, insightful comments
- The Introduction is quite short. There is a good deal of research on this subject that is not cited. I'd suggest that another 10-12 references be included. Some attention may be given to the role of social supports on PPD (embeddedness in networks such as extended family, community groups, congregations, etc.). Even if the influence of these support networks cannot be measured due to data limitations (see comment 2), this is worth mentioning at the outset.
2-3p: Recent meta-analyses have found that PPD is influenced by various demographic, medical, psychological, social, and instrumental factors [3,4]. Demographic factors, including young maternal age, lower education levels, low socioeconomic status, and unemployment, increase the risk of PPD [3,4]. Medical factors, such as gestational diabetes, pregnancy complications, cesarean delivery, the use of epidural anesthesia, and infant health issues (e.g., low birth weight, intensive care unit admission), are also significant predictors [3,4]. Psychological risk factors, including history of depression or anxiety, high neuroticism, emotional instability, and trauma, further contribute to PPD vulnerability [3,4]. Among social factors, inadequate support from family [3], female friends, or one’s mother [6] has been reported to increase PPD risk. Lastly, instrumental factors, including limited healthcare access, lack of mental health screening, and unhealthy lifestyles, further exacerbate the risk of PPD [3,4].
However, the most influential factor remains unclear because several studies have reported different variables. For example, one study reported that the husband’s low level of education, low social support, vacuum use during delivery, and stressful life events during pregnancy were the most significant factors affecting PPD [5]. Another study identified a wide range of relevant factors, including immigration challenges, obesity or dissatisfaction with body image, vitamin D deficiency, genetic and epigenetic markers, low self-esteem, multiple births, postpartum anemia, cultural preferences for male children, and social stigma surrounding mental health [7].
Machine learning techniques may help address these challenges by identifying significant features that contribute to predicting health outcomes. By integrating a multitude of parameters, machine learning can offer a more objective assessment of PPD risk, potentially enabling early interventions and improved outcomes [8]. Machine learning models have been applied to predict PPD using diverse datasets, including electronic health records (EHRs) and patient-reported outcomes. These models identified key predictive factors across demographic (age, marital status, and socioeconomic status) [9-12], medical (pregnancy complications, body mass index, and medication) [9,13-16], psychological (neuroticism, resilience, emotional stability) [10,11,17], social (number of people living together) [18], and instrumental (healthcare access and prenatal care) [12,15,17] domains. Among these, mental health history, pregnancy complications, and socioeconomic conditions were the strongest predictors of PPD risk [9,10,14,15]. Machine learning models have highlighted the significant influence of demographic and medical factors in predicting the risk of PPD. However, there remains a lack of machine learning analysis of social aspects, individual emotional characteristics, and health behaviors.
- Does the data set feature social support variables such as those listed in my foregoing comment? That is a missing component here. If not, please circle back to that limitation in the Discussion and recommend that as a path for future research.
15p: Another limitation was related to the psychological predictors used in the PPD. The analysis focused on stress, conflict with a partner, and the value of children, but did not include other significant psychosocial factors. Key predictors such as emotional resilience, coping mechanisms, personality traits (e.g., neuroticism), and perceived social support were absent, limiting the model's comprehensiveness.
- A Methods section of one sentence seems really limited. Could this section be expanded to at least 2-3 paragraphs that delve more deeply into the methods applied and their value? This is an opportunity to make a strong case for your design and it is a missed opportunity. As one example, you state the following in the Introduction: "By integrating a multitude of parameters, machine learning can offer a more objective assessment of PPD risk, potentially enabling early interventions and improved outcomes [6]." A statement like this could be expanded on in the Methods section. How does machine learning offer a more objective assessment? Would this approach expected to be superior to more conventional methods such as regression? Unpack the logic and potential advantages of machine learning approaches for those unfamiliar with them. Moreover, since there are multiple machine learning approaches, detailing those variations and their relative potential contributions seems appropriate here. One sentence is simply too terse.
4-5p: The cross-sectional approach facilitated a snapshot analysis of key associations with PPD in a large, representative sample of postpartum women. In large samples, machine learning serves as a powerful computational tool for analyzing multivariate medical data [8].
Machine learning was applied in this study as an advanced analytical tool to improve the accuracy of PPD risk prediction, an area where traditional statistical models have limitations. Unlike classical regression techniques, which rely on predefined assumptions about the relationships among variables, machine learning models can analyze complex, non-linear interactions among multiple predictors, uncovering hidden patterns that may not be apparent in traditional analyses. The use of algorithms such as random forest, AdaBoost, and logistic regression enabled a comparative assessment of classification performance, ensuring that the most effective model was selected based on empirical metrics such as accuracy, precision, recall, and area under the receiver operating characteristic curve (AUC-ROC) scores.
Machine learning logistic regression and general logistic regression differ in purpose, methodology, and applications. General logistic regression is primarily used for inferential analysis and hypothesis testing, focusing on understanding relationships between variables. In contrast, machine learning logistic regression is designed to optimize performance in predictive modeling and classification. Additionally, machine learning logistic regression scales better for high-dimensional and large datasets, whereas general logistic regression struggles with complexity and overfitting [18].
- I have always thought that the Value for Children scale should be renamed the Value of Children scale, but I expect no name change here. You might add that it is sometimes called the Children's Perceived Value scale. It is my understanding that these names are used interchangeably.
Value of Children
- Breastfeeding should be one word. It is sometimes one word (text) and other times two words (tables).
Breastfeeding
- I'd welcome the addition of citations to support the use of SMOTE, AUC-ROC, and any analytical approach employed here. I will readily admit that I am not an expert on these methods. Yet, their use should be justified with citations to relevant peer-reviewed publications. Be sure that each is sufficiently explained without presuming prior knowledge on the part of the reader.
9p: ROC-AUC is a performance metric for binary classification models that evaluates their ability to distinguish between classes. Accuracy is the proportion of correctly classified instances out of the total predictions, providing a general measure of a model’s performance. Precision refers to the proportion of correctly predicted positive instances among all predicted positives, which is particularly important in cases where false positives are costly. Recall represents the proportion of actual positive instances correctly identified by the model, which is crucial when minimizing false negatives is a priority. The F1-score is the harmonic mean of precision and recall, serving as a balanced metric for evaluating models, especially in imbalanced datasets where both false positives and false negatives need to be considered [29].
- If I understand correctly, the author used a cross-section of data from a panel study. We cannot then say that Value for Children has an effect or influence on PPD. The causal direction here is impossible to determine from a single wave of data. Value for Children is a dynamic construct that could influence or by influenced by PPD. So, an argument that Value for Children "influences the occurrence of PPD" should be "is associated with the occurrence of PPD." Special care should be taken to re-read the entire manuscript with causal order considerations and data limitations in mind. If the data analyzed here are longitudinal, please state that more clearly.
1p: Conflict with a partner and stress levels emerged as the strongest predictors. Higher levels of conflict and stress were associated with an increased likelihood of PPD, whereas a higher value of children reduced this risk. Maternal psychological status and environmental features should be managed carefully during the postpartum period.
12p: In addition, the study found that the value of children was significantly related to the occurrence of PPD. This represents a novel analytical finding that has rarely been explored in previous machine learning–based PPD studies [31].
14p: Future research should further explore these associations across diverse populations and cultural contexts to better understand how social and psychological factors interact in influencing PPD risk [6].
- What broader conclusions about machine-learning approaches versus regression models can be discerned from this manuscript?
16-17p: This study provides important insights into the relationship between partner social support and PPD, an area that has been insufficiently addressed in both classical statistical and machine learning analyses. The findings indicate that partner-related factors, including conflict with a partner and paternal involvement, are significantly associated with PPD risk. Higher levels of partner conflict were linked to an increased likelihood of PPD, while greater paternal involvement was associated with lower depressive symptoms. These results suggest that the quality of partner support during the perinatal period plays a crucial role in maternal mental health. Additionally, this study highlights a novel association between maternal social values and beliefs about children and PPD risk. Mothers who placed a higher value on children exhibited lower levels of PPD symptoms, suggesting that positive perceptions of motherhood may have a protective effect on mental well-being. This finding emphasizes the potential role of maternal attitudes and societal beliefs in shaping postpartum emotional experiences.
This is a promising study but needs considerable additional work to realize its potential.
I sincerely thank the reviewer for your valuable time, insightful comments, and constructive suggestions, which have greatly improved the quality of this manuscript. I appreciate your thoughtful review and the opportunity to strengthen our work based on your feedback.
Round 2
Reviewer 3 Report
Comments and Suggestions for Authors I support the study moving forward with acceptance.